# Effects of Varying Dietary Concentrations of Menadione Nicotinamide Bisulphite (VK_3_) on Growth Performance, Muscle Composition, Liver and Muscle Menaquinone-4 Concentration, and Antioxidant Capacities of Coho Salmon (*Oncorhynchus kisutch*) Alevins

**DOI:** 10.3390/biology14040447

**Published:** 2025-04-20

**Authors:** Han Zhang, Leyong Yu, Abdur Rahman, Sattanathan Govindharajan, Lingyao Li, Hairui Yu, Muhammad Waqas

**Affiliations:** 1College of Fisheries and Life Science, Shanghai Ocean University, Shanghai 201306, China; a787382102@gmail.com; 2Key Laboratory of Biochemistry and Molecular Biology in Universities of Shandong, Weifang Key Laboratory of Coho Salmon Culturing Facility Engineering, Institute of Modern Facility Fisheries, College of Biology and Oceanography, Weifang University, Weifang 261061, China; leyong618@gmail.com (L.Y.); sattanathanphd@gmail.com (S.G.); 980714742@163.com (L.L.); 3Shandong Freshwater Fisheries Research Institute, Jinan 250013, China; 4College of Fisheries, Huazhong Agricultural University, Wuhan 430070, China; 5Weifang Key Laboratory of Salmon and Trout Health Culture, Conqueren Leading Fresh Science & Technology Inc., Ltd., Weifang 261205, China; 6Department of Animal Sciences, University of Veterinary and Animal Sciences, Lahore 54000, Pakistan; 7Shandong Collaborative Innovation Center of Coho Salmon Health Culture Engineering Technology, Shandong Conqueren Marine Technology Co., Ltd., Weifang 261108, China; 8Department of Animal Nutrition and Nutritional Diseases, Faculty of Veterinary Medicine, Ondokuz Mayıs University, Samsun 55139, Türkiye; m.waqas@uvas.edu.pk

**Keywords:** antioxidant, aquaculture, menadione, menaquinones, vitamin, growth

## Abstract

Coho salmon farming has experienced rapid expansion in China, driven by the species’ fast growth, exceptional nutritional value, and highly desirable fillet quality. Rich in high-quality protein, omega-3 fatty acids, and carotenoids such as astaxanthin, coho salmon is widely recognized for its health benefits, making it a valuable commodity in both domestic and international markets. However, the intensification of large-scale farming has created a pressing need for optimized feed formulations to ensure sustainable production and maintain fish health. Despite its economic significance, the specific nutritional requirements of coho salmon remain insufficiently explored, particularly regarding vitamin K_3_ (menadione), an essential micronutrient that plays a pivotal role in blood coagulation, bone metabolism, antioxidant activity, and overall growth. To bridge this critical knowledge gap, optimizing the dietary dosage of VK_3_ for coho salmon alevins is imperative. This study focuses on the optimization of the dosage of vitamin K_3_ and its beneficial effects on growth performance, muscle composition, and antioxidant capacity during freshwater rearing.

## 1. Introduction

Fish is a vital part of the human diet and is well known for its health benefits. Its global consumption has nearly doubled over the last four decades [1,2]. In fish farming, the provision of complete balanced feed, including macronutrients along with micronutrients, is the fundamental nutritional strategy to obtain the maximum yield by avoiding health-related issues [3] and skeletal anomalies during the developmental phase [4,5,6]. Although the micronutrients are equally and synergistically involved in imparting the real-time effects of all nutrients into the final yield, the majority of the research has been focused on exploring the requirements of macronutrients in fish farming, while the assessment of micronutrients for growth and allied parameters has remained neglected [7]. Especially with the new trends of intensive freshwater fish farming, it is becoming imperative to optimize and fulfill the need for all nutrients for cost-effective and efficient production of fish during early growth stages [8]. The early growth stage is critical for nutritionists to provide varying levels of required nutrients. The requirements of the early development stage are usually different than the subsequent life periods and proper care during this phase determines the future life span success [9,10,11]. Among the various micronutrients, vitamin K3 (VK3), a fat-soluble compound, has received particular attention in fish feed, especially in the current context of freshwater aquaculture [12]. Based on its origin, vitamin K has different forms: phylloquinone (VK1) is derived from plants, menaquinone (VK_2_) is derived from microbes, and menadione (VK_3_) is the synthetic provitamin form of vitamin K_3_ [13,14]. The naturally occurring VK in the feeds is sometimes not sufficiently available, and synthetic derivatives of VK_3_, especially menadione (VK_3_), are supplemented in the feeds [15,16], which is converted into menaquinone-4 (MK-4) to be available in tissue metabolism [17]. Vitamin K is a micronutrient and is needed in minute quantities for various vital biochemical and physiological functions in the body. It promotes growth and blood coagulation in case of injury by producing thrombin [18]; it is also helpful in bone formation and mineralization by osteoblast cells through the production of osteocalcin, a calcium-binding protein for mineralization of the bone matrix; and it regulates the homeostasis of calcium and bone, as well [18,19]. The mechanism of involvement of VK_3_ in this process is through the protein carboxylation with the help of MK-4 as a cofactor along with γ-glutamyl carboxylase to convert glutamyl residues to produce clotting factors, osteocalcin, and Gla matrix for bone mineralization [19,20,21]. The presence of VK_3_ in an optimal concentration in the fish diet is highly essential to maintain growth, homeostasis, and bone strength [8], and deficiency of VK_3_ will lead to skeletal disorders, anemia, prolonged clotting time, and may ultimately lead to poor growth [22,23,24,25,26].

Despite the known fact that the optimal dietary concentration of VK_3_ in feed is an essential nutrient to achieve normal growth, bone development, and overall health status during early life, no or little literature is available on the dose optimization of VK_3_ in coho salmon alevins in their early stages of development. However, in some research studies, it is evident that VK_3_ addition to the diet has improved the growth performance, nutrient digestion, and antioxidant status in Jian juvenile carp fish (*Cyprinus carpio)* [26] with a dose rate of 3.13 mg/kg of diet, and in Senegalese sole (*Solea senegalesis*) larva up to 250 mg/kg [8]. Some other studies demonstrated that vitamin K supplementation in the diet has improved feed efficiency and protein efficiency ratio in fleshy prawn (*Penaeus chinensis*) shrimp [27]; antioxidant parameters were improved in *Haliotis discus hannai* (abalone) [28]; improved weight and feed efficiency were observed in gilthead seabream (*Sparus aurata*) fingerlings [16]; 5 mg/kg dietary vitamin K has resulted in better growth and bone development in gilthead seabream larva [29]; and dietary vitamin K_3_ does not affect growth but improves antioxidant status and immunity in whiteleg shrimp (*Litopenaeus vannamei)* [30]. On the other hand, some studies claimed that there is no need for dietary supplementation of VK_3_ and naturally occurring VK_3_ in basal feed is sufficient for growth, but an optimal dietary concentration of 2–4 mg/kg is required for other body functions in Nile Tilapia (*Oreochromis niloticus*) juveniles [31]. Dai et al. [30] reported that dietary VK_3_ had no effect on growth, but 39.06 mg/kg improved antioxidant status and immunity in whiteleg shrimp. The nutrient requirement for VK varies with age. The VK requirements for juvenile and adult fish in some species including grass carp (*Ctenopharyngodon Idella*), lake trout (*Salvelinus namaychus*), Mummichog (*Fundulus heteroclitus), Pseudosciaena crocea* large yellow croaker (*Larimichtys crocea* juveniles, Jian carp juveniles, largemouth bass (*Micropterus salmoides*), and gilthead seabream fingerlings [15,16,24,32,33,34] have been quantified, but no quantifications of dietary vitamin K_3_ requirements for the larval stage of coho salmon (*Oncorhynchus kisutch)* has been conducted [8]. This literature gap urges us to investigate the need, importance, and dose optimization of VK_3_ in coho salmon alevins.

Coho salmon, a species of significant commercial importance, is anadromous, meaning it migrates between freshwater and marine environments throughout its life cycle, from hatching to spawning. [35,36]. Coho salmon is highly regarded for its exceptional nutritional profile, fast growth, delicious meat, high-quality protein and polyunsaturated fatty acids (PUFAs), including significant levels of omega-3 fatty acids. Additionally, it contains astaxanthin and phospholipids, which are well known for their health benefits, including antioxidant, anti-inflammatory, anti-thrombotic, and cardio-protective properties [37,38,39]. Due to its high nutritional value and economic potential, coho salmon farming has expanded rapidly worldwide, particularly in China [40]. This expansion of large-scale farming of coho salmon has increased the demand for optimized feed formulations to support sustainable production. However, research on the nutritional requirements of coho salmon remains limited, particularly regarding dietary vitamin K, which plays a crucial role in physiological processes such as blood coagulation, bone metabolism, antioxidant activity, and overall growth [41]. Addressing this gap in the literature, it was hypothesized that optimizing the dietary vitamin K3 (menadione) dosage for coho salmon alevins is crucial to enhance growth performance, improve muscle composition, and support antioxidant capacity during freshwater rearing.

Therefore, the present study was designed to evaluate the effects of different dietary vitamin K_3_ inclusion levels on growth performance, body muscle composition, MK-4 concentration, and antioxidant capacity in coho salmon alevins. The findings of this study will contribute to a better understanding of coho salmon nutrition and provide valuable insights for optimizing feed formulation in aquaculture.

## 2. Materials and Methods

### 2.1. Ethical Statement

The present study was carried out according to the recommendations (animal handling, care, and housing) in the “Guide for the Ethical Use of Experimental Animals” of the Weifang University, Weifang, Shandong, China (No. 20200601) and it was approved by the Institutional Animal Care and Use Committee of the Institute of Modern Facility Fisheries, Weifang University, China (IACUC NO. 20220516003).

### 2.2. Experimental Diets

Menadione nicotinamide bisulphite (VK_3_) was procured from Shanghai Yuanye Bio-Technology Co., Ltd., Shanghai, China. Seven isocaloric and isonitrogenous diets were formulated with supplemented levels of VK_3_: 0.00; 5.00, 10.00, 15.00, 20.00, 40.00, and 60.00 mg/kg VK_3_, producing analyzed dietary VK_3_ contents of 0.16, 5.25, 10.22, 14.93, 20.51, 40.09, and 59.87 mg/kg, respectively. The ingredients and chemical composition of diets are given in Table 1.

### 2.3. Animals and Feeding Management

The coho salmon alevins were procured from the hatchery of the National Aquatic Product Introduction and Breeding Centre (Beijing, China). The coho salmon alevins were cultured and reared at one of the experimental base sites of the Coho Salmon Health Culture Engineering Technology Center of Shandong Collaborative Innovation Center (Weifang, China). A total of 2100 alevins were divided into 21 tanks, with 100 coho salmon alvenis in each tank, and 3 tanks represented one group. The fish were fed in 21 white plastic tanks (0.8 × 0.6 × 0.6 m, L × W × H, and water volume 240 L/tank) with 100 fish per tank in triplicate per dietary treatment. There were seven treatment groups. Fish were fed four times daily (7:00, 10:30, 14:00, and 17:30) for a trial period of 12 weeks. The feeding rate was 3% of the body weight. The alevins were reared in filtered underground spring water and natural light during the feeding period. The temperature, dissolved oxygen, and pH were in the range of 15.7 ± 0.5 °C, 9.3 ± 0.5 mg/L, and 7.1 ± 0.2, respectively.

### 2.4. Euthanasia Methods for Coho Salmon Alevins

At the end of the feeding trial, the fish were withheld from feed for 24 h, anesthetized with tricaine methanesulfonate (MS-222; Wuhan Biocar Pharmaceutical Co., Ltd., Wuhan, China) at a concentration of 30 mg/L, and then quickly frozen using liquid nitrogen to −80 °C just after finishing the samples at tank sites and immediately transferred to the laboratory and then stored at −20 °C in the refrigerator for the subsequent analyses.

### 2.5. Sampling Procedures and Analytical Methods

#### 2.5.1. Growth Performance

Body weight and number of fish counts were recorded for all fish in each replicate at the start and end of the trial. Then, at the end of the trial, three fish from each replicate were randomly caught, and their final weight and body length were recorded. These fish were then immediately dissected to collect the whole-body muscle, liver, and viscera samples. The whole body, liver, and viscera of the fish were weighed and stored at −20 °C for further analysis. Fish muscle samples were used for muscle composition analysis and whole-body MK-4 concentrations. Liver samples were used to assess the MK-4 concentrations and total antioxidant capacity (T-AOC). The above-recorded values about growth performance were then used to calculate the survival rate (SR), weight gain (WG), specific growth rate (SGR), feed conversion ratio (FCR), Fulton condition factor (K), hepatosomatic index (HSI), and viscerosomatic index (VSI) by using the equations below [42].SR (%) = (final fish number/initial fish number) × 100WG (g) = final weight (g) − initial weight (g)SGR (%/d) = 100 × ((lnfinal body weight) − (lninitial body weight))/daysFCR = total feed intake (g)/(final body weight (g)-initial body weight (g))Protein intake (g) = total feed intake (g) × protein percent in feedK (%) = (W/l^3^) × 100; [(W = weight (g); Tl = total length (cm^3^)]HSI (%) = (liver weight/final weight) × 100VSI (%) = (visceral weight/final weight) × 100

#### 2.5.2. Analysis of Feed Samples and Chemical Composition of Muscles

The feed samples and stored muscle and liver samples from the dissected fish were analyzed by the methods prescribed in AOAC [43] for nutrient composition analysis; crude protein was determined by Kjeldhal apparatus (Automatic Apparatus Kjeltec 8400, FOSS, Hillerød, Denmark) by calculating nitrogen (N × 6.25); crude fat was determined by Soxhlet apparatus (ST 243, FOSS, Hillerød, Denmark); moisture was determined at 105 °C for four h in a hot air oven to constant weight; and samples were burnt in a muffle furnace at 550 °C for 12 h for crude ash analysis.

#### 2.5.3. MK-4 Concentration Determination in Whole-Body Muscles and Liver Tissue, and Liver Anti-Oxidative Enzyme Activity Analysis

The whole-body muscle (1.0 g) and liver (0.2 g) samples were separately homogenized. The homogenate of the muscle sample and liver were then analyzed by following the technique prescribed by Ostermeyer and Schmidt [44] for whole-body muscle and liver menaquinone-4 (MK-4) concentration analysis.

The liver samples were homogenized and placed in 0.1 M pH 7.4 Tris-HCl buffer (Qingdao Miruida Trade Co., Ltd., Qingdao, China) at 4 °C, and then used for analysis. The liver anti-oxidative enzyme activity, including total antioxidant capacity (T-AOC), total superoxide dismutase (T-SOD) activity, catalase (CAT), and liver malondialdehyde (MDA) content were determined by the commercial reagent kit (Nanjing Jiancheng Bioengineering Institute, Nanjing, China) with the help of an automatic analyzer.

#### 2.5.4. Statistical Analysis

The data obtained from the experiment were statistically analyzed by one-way ANOVA in SPSS software (IBM SPSS 21.0) and the mean value was tested by the Duncan test. The values in the table are presented as mean ± SE and the level of significant difference was *p* ≤ 0.05. The polynomial quadratic regression was used (Microsoft Excel 97-2003 Worksheet and Chart Design) on the data of WG, SGR, liver vitamin K (MK-4), and T-AOC to predict the optimal required dietary concentration of VK_3_ for these parameters in coho salmon alevins.

## 3. Results

### 3.1. Growth Performance

Growth performance parameters have been presented in Table 2. The survival rate of the coho salmon alevins was recorded significantly higher in all dietary treatments as compared to the control basal diet (0.16 mg/kg). Among the treatments, the highest survival rate (*p* < 0.05) was in diet groups supplemented with 59.87 and 20.51 mg/kg of VK_3_ followed by 40.09, 14.93, 10.22, and 5.25 mg/kg diet groups, while the control (0.16 mg/kg) group was observed with the lowest survival rate. Weight gain and SGR were observed to be higher (*p* < 0.05) in all VK_3_-supplemented groups as compared to the control group. The highest weight gain was observed in groups with 14.93 and 20.51 mg/kg VK_3,_ and the highest SGR was in the diet group of 14.93 mg/kg VK exclusively. The FCR was also found to be better (*p* < 0.05) in all VK_3_-supplemented groups as compared to the control group, with pronounced better effects in the diet group with 14.93 mg/kg and 20.51 mg/kg of VK_3_, followed by the 40.09 and 59.87 mg/kg dose groups. The other parameters, including K, HSI, and VSI, remained unaffected (*p* > 0.05) in all of the treatment groups (Table 2). The predicted optimal required concentration of VK_3_ for maximum WG and SGR was found to be 34 mg/kg and 43.50 mg/kg, respectively, by polynomial quadratic regression analysis in the coho salmon alevin diet (Figure 1 and Figure 2).

### 3.2. Muscle Chemical Composition

Muscle chemical composition analysis revealed no effect (*p* > 0.05) on muscle composition, including moisture percentage, crude protein, crude lipid, and/or ash content with any level of VK_3_ supplementation (Table 3).

### 3.3. MK-4 Concentrations in Whole-Body and Liver Tissue and Liver Anti-Oxidative Enzyme Activity

Whole-body MK-4 and whole-liver MK-4 concentrations were recorded to be higher (*p* < 0.05) in all diets with high VK_3_ dietary levels as compared to the control (0.16 mg/kg of VK_3_). The highest value of whole-body MK-4 was noticed in the 59.87 mg/kg VK_3_ dietary group, and the highest liver MK-4 values were recorded in the 59.87, 40.09, and 20.51 mg/kg of VK_3_ dietary groups (Table 4). Quadratic regression analysis predicted that 38.54 mg/kg was a suitable required dietary concentration of VK_3_ to achieve maximum liver vitamin K (MK-4) concentration in coho salmon alevins (Figure 3).

Whole-liver T-AOC, T-SOD, and CAT were significantly higher (*p* < 0.05) in all groups receiving higher dietary treatment levels compared to the control group. The highest values of T-AOC were in diet groups 10.22, 14.93, 20.51, and 40.09 mg/kg of VK_3_, while in the case of T-SOD and CAT, the highest values were found in the dietary group 14.93 mg/kg VK_3_. The predicted required concentration of dietary VK_3_ for liver T-AOC was found to ebb to 31.97 mg/kg in coho salmon alevins (Figure 4). In the case of MDA contents, the results revealed significantly lower values (*p* < 0.05) in all dietary levels as compared to the control. The significantly lowest (*p* < 0.05) values of MDA contents were found in the dietary group with 14.93 and 20.51 mg/kg VK_3_ (Table 5).

## 4. Discussion

It is well documented that Vitamin K is essential in attaining and maintaining optimal growth during early growing and developmental stages [8,34]. The deficiency and excessive intake of VK_3_ have also been reported to reduce growth performance [27,45,46].

In the current study, results showed that VK_3_ addition to the diet has improved the SR, WG, SGR, and FCR in all VK_3_-supplemented groups. Different researchers have studied different doses for growth parameters. In line with the present study, an increase in body weight gain was reported at 6.5 to 9.8 mg/kg menadione in cod, while suggesting a decrease in body weight on excess or low dietary inclusion of VK_3_ (43). Wei et al. [34] recommended adding 15.08 mg/kg menadione sodium bisulfite to the diet of largemouth bass, but negative WG and SGR were observed with 20.85 mg/kg, pointing out toxicity effects at higher doses. Similarly, Sivagurunathan et al. [29] have found that the addition of VK_3_ up to 70 mg/kg has shown improvement in growth and survival rate in gilthead seabream larvae. Additionally, Dominguez et al. [16] recommended the use of an optimal level of 12 mg/kg of VK_3_ in gilthead seabream fingerlings for improvement in growth, while similar recommendations for growth with increasing levels of VK_3_ were suggested by Richard et al. [8] in Senegalese sole larvae. In contrast to the present study, some researchers found no effect of VK_3_ addition in the diet on growth-related parameters. For example, Abdelhamid et al. [31] found no effect in juvenile Nile tilapia on the addition of 2–12 mg/kg VK_3_, and Dai et al. [30] found that 9.97–156.09 mg/kg of VK_3_ had no effect in shrimps, while Grisdale-Helland et al. [47] suggested avoiding the use of menadione sodium bisulfite (MSB), noting that vitamin K_1_ was more effective in reducing mortality in Atlantic salmon (*Salmo salar*). They suggested that MSB may increase the fish’s susceptibility to environmental and pathogenic stressors, possibly due to its lower stability and higher toxicity, as well as its tendency to undergo redox reactions that generate reactive oxygen species, leading to oxidative stress and cytotoxicity. The variation in VK_3_ requirements observed across studies may be attributed to different fish species, growth stages, rearing conditions, and feeding nature. Based on the findings of our research regarding growth parameters, we concluded that the predicted required dietary concentration of VK_3_ for WG and SGR was 34.0 and 43.50 mg/kg, respectively, for coho salmon alevins during early growth.

The dietary ingredients have a direct impact on the muscle composition [48]. The different concentrations of VK_3_ in the diet have revealed no effect on the whole-body chemical composition during the present study. Likewise, Wei et al. [34] found no effect of VK_3_ dietary supplementation on body muscle composition, including crude protein, crude fat, ash, and moisture contents up to 20.85 mg/kg VK_3_ in largemouth bass. Dominguez et al. [16] also stated that whole-muscle composition was not altered with dietary VK_3_ addition in the diet of gilthead seabream fingerlings. Graff et al. [49] also supported this claim in Atlantic salmon. This indicates that the presence of VK_3_ in the diet has no adverse effects on the muscle composition.

The VK_3_ acts as a pro-vitamin K and is converted to menaquinone-4 (MK-4) in body tissues [45] and stored mainly in the liver [23]. This study’s results showed that during whole-body muscle and liver sample analyses for MK-4, the concentrations of MK-4 were increased with a gradual increase in dietary VK_3_ supplementation. Similar findings were noticed by Grahl-Madsen and Lie [45] in Atlantic cod (*Gadus morhua*) with a dietary concentration of up to 9.93 to 15.22 mg/kg VK_3_ in the diet. Wei et al. [34] also endorsed that the addition of VK_3_ in the diet may increase the MK-4 in the liver of largemouth bass with the optimal dietary concentration of 11.20 mg/kg VK_3_. Vera et al. [50] added that increasing the level of dietary VK_3_ has an increasing effect on the whole-body MK-4 concentration in adult Atlantic salmon, and this was also supported by Graff et al. [49] in Atlantic salmon parr. However, no supporting literature was available for adding and quantifying the dietary concentration level of VK_3_ in coho salmon alevins. However, contrary to the findings of this study, previous research [34] suggests that higher doses (20.85 mg/kg) of MNB may lead to a decline in MK-4 concentration in largemouth bass due to potential toxicity effects. In the present study, dietary VK₃ was efficiently converted into menaquinone-4 (MK-4) without any adverse effects on liver metabolism, and MK-4 was distributed throughout the body as dietary VK₃ levels increased. The optimal dietary requirement of VK₃ for achieving adequate MK-4 levels in the liver was estimated to be 38.54 mg/kg. The higher MK-4 concentrations observed in both the liver and the whole body suggest that increasing VK₃ doses may provide more substrate for conversion to MK-4, as lower VK₃ may degrade during the conversion process or feed manufacturing, leading to a reduced bioavailable amount of VK₃ for conversion into MK-4 in the tissues [17]. Therefore, the increase in dietary VK₃ likely resulted in higher availability of MK-4 in both the liver and body. Regardless of the other physiological effects of VK_3_ in fish, it protects the body by reducing the production of peroxides, as these peroxides are mainly involved in lipid peroxidation by producing free radicals [31,51,52]. The liver enzymes are present in the liver and are released into the blood when damage to cells occurs. The high concentration of these liver enzymes will represent higher values in the blood too. Antioxidant enzymes, such as superoxide dismutase (SOD) and catalase (CAT), act as natural scavengers of free radicals. They protect cells from oxidative stress and prevent lipid peroxidation caused by reactive oxygen species (ROS), thereby enhancing the overall antioxidant status [31,32,53]. Total antioxidant capacity (T-AOC) serves as a comprehensive indicator of the body’s antioxidant defense system. It plays a crucial role in protecting the body against oxidative damage induced by free radicals [54].

Certain natural bioactive compounds, including VK_3_, have the capacity to increase the production of SOD and CAT, which ultimately help to capture the free radicals, thus improving the antioxidant capacity in the fish at 39.06 mg/kg VK_3_ in the diet [30], and up to 2–4 mg/kg was reported by Abdelhamid et al. [31]. Similarly, MDA is the product of lipid peroxidation, which increases with more free radical reactions and is considered a potent indicator of tissue damage in the liver [55]. Some nutraceuticals have shown the potential to reduce lipid peroxidation by various mechanisms, including the increase in the concentration of liver SOD and CAT. In the current study, the increasing concentration of dietary VK_3_ has increased the T-AOC, T-SOD, and CAT while it has decreased the concentration of MDA in liver samples, indicating the positive effects of VK_3_ on liver health [32]. In line with the present study, Dai et al. [30] reported that the addition of 39.06 mg/kg VK_3_ in the diet has increased the T-AOC and catalase enzyme concentration in serum while it has reduced the MDA contents in whiteleg shrimp. Similarly, Wei et al. [34] also reported that the optimal dose of up to 9.93 to 15.22 mg/kg VK_3_ in the diet of largemouth bass was found to be effective to improve SOD and T-AOC contents and to lower MDA contents in the liverQuadratic regression analysis in the present study has predicted the optimal required dietary VK_3_ level of 31.97 mg/kg for better T-AOC in coho salmon alevins.

## 5. Conclusions

In conclusion, this study demonstrates that the inclusion of synthetic vitamin K_3_ has significantly improved the growth performance, whole-body MK-4, and liver MK-4 concentrations and liver total antioxidant capacity in coho salmon alevins at a dietary concentration of above 0.16 mg/kg in all treatments. Quadratic regression analysis based on the WG, SGR, liver MK-4, and liver T-AOC predicted the optimal dietary vitamin K_3_ levels to be 34.00, 43.50, 38.54, and 31.97 mg/kg, respectively, in coho salmon alevins.

## Figures and Tables

**Figure 1 biology-14-00447-f001:**
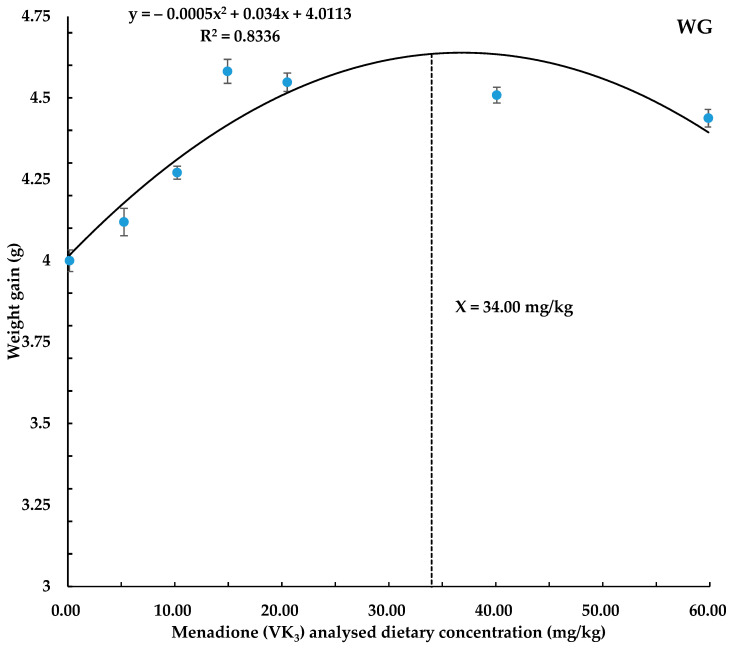
Polynomial quadratic regression analysis of WG with graded levels of vitamin K_3_ in coho salmon alevins predicted that the optimum vitamin K_3_ requirement for WG was 34 mg/kg. Abbreviation: WG, weight gain; dotted line shows X value and complete curve line shows trend.

**Figure 2 biology-14-00447-f002:**
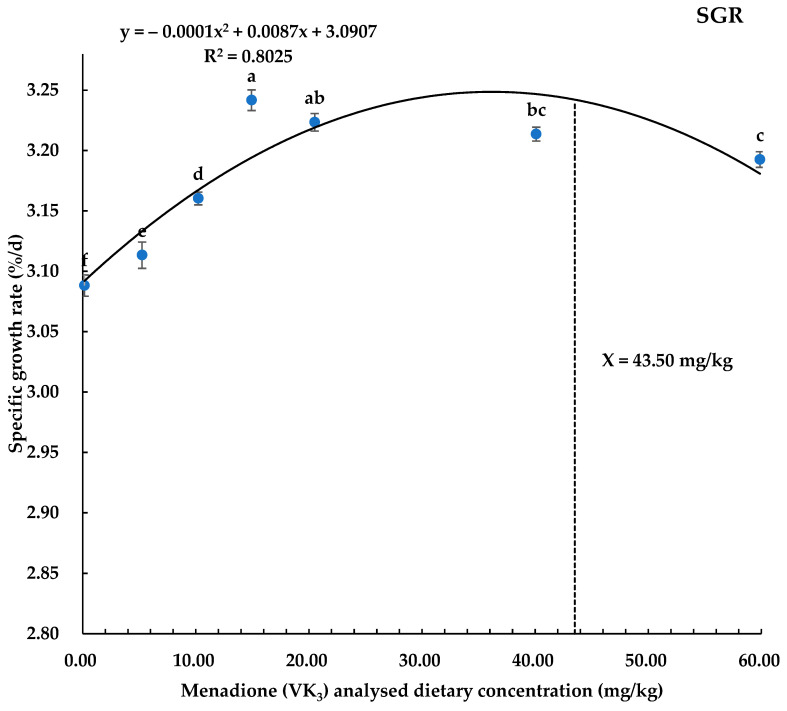
Polynomial quadratic regression analysis of SGR with graded levels of vitamin K_3_ in coho salmon alevins predicted that the optimum vitamin K_3_ requirement for SGR was 43.50 mg/kg. Abbreviation: SGR, specific growth rate; dotted line shows X value and complete curve line shows trend.

**Figure 3 biology-14-00447-f003:**
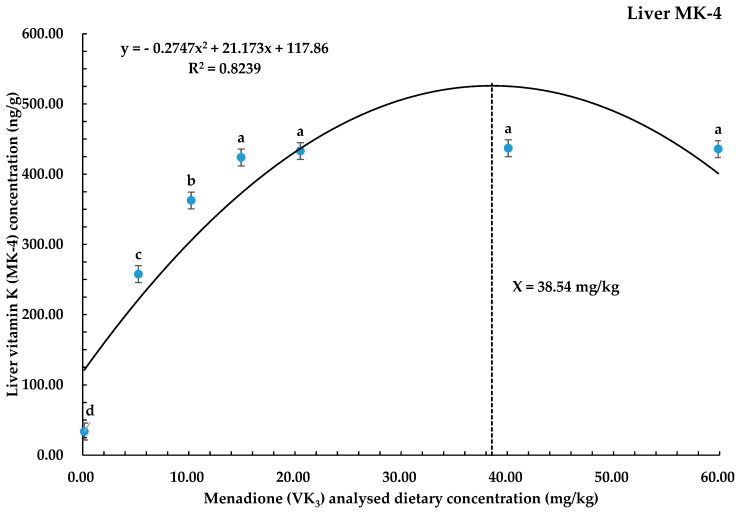
Polynomial quadratic regression analysis of liver vitamin K (MK-4) with graded dietary vitamin K_3_ levels for coho salmon alevins predicted that the optimum vitamin K_3_ requirement for liver MK-4 was 38.54 mg/kg. Abbreviations: MK-4, menaquinone-4; dotted line shows X value and complete curve line shows trend.

**Figure 4 biology-14-00447-f004:**
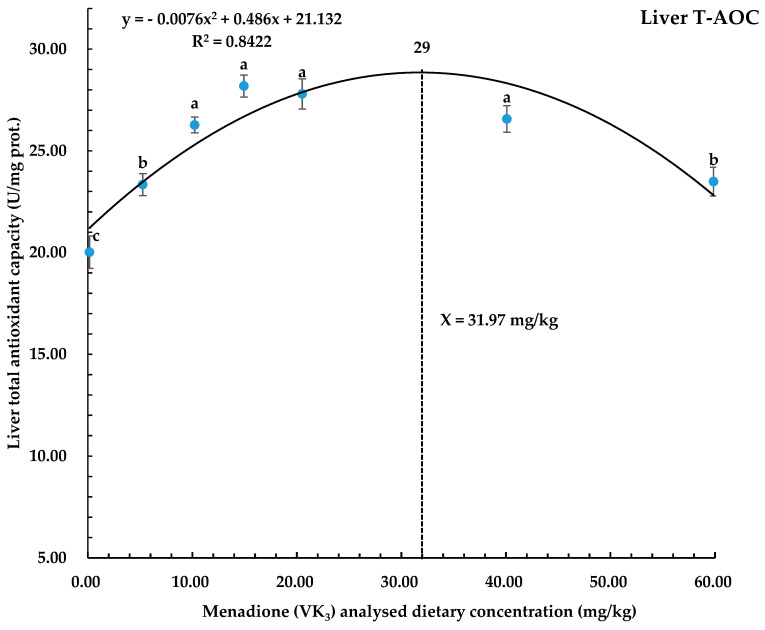
Polynomial quadratic regression analysis of T-AOC with graded levels of vitamin K_3_ in coho salmon alevins predicted that the optimum vitamin K_3_ requirement for T-AOC was 31.97 mg/kg. Abbreviations: T-AOC, total antioxidant capacity; dotted line shows X value and complete curve line shows trend.

**Table 1 biology-14-00447-t001:** Formulation and proximate composition of the experimental diets for coho salmon alevins (% dry matter).

Ingredients	Dietary Vitamin K_3_ Levels (mg/kg)						
	0.16	5.25	10.22	14.93	20.51	40.09	59.87
Casein ^1^	38.00	38.00	38.00	38.00	38.00	38.00	38.00
Gelatin ^1^	12.00	12.00	12.00	12.00	12.00	12.00	12.00
Dextrin ^1^	28.00	28.00	28.00	28.00	28.00	28.00	28.00
Corn oil ^1^	6.00	6.00	6.00	6.00	6.00	6.00	6.00
Fish oil ^1^	3.00	3.00	3.00	3.00	3.00	3.00	3.00
α-cellulose ^1^	8.00	8.00	8.00	8.00	8.00	8.00	8.00
Mono-calcium phosphate ^1^	1.50	1.50	1.50	1.50	1.50	1.50	1.50
Mineral premix ^2^	2.50	2.50	2.50	2.50	2.50	2.50	2.50
Vitamin premix (vitamin K_3_ free) ^3^	1.00	1.00	1.00	1.00	1.00	1.00	1.00
Vitamin K_3_ (mg/kg)	0.00	5.00	10.00	15.00	20.00	40.00	60.00
Proximate nutrient composition							
Moisture	10.37	10.28	10.49	10.22	9.91	10.44	10.31
Crude protein	45.12	45.15	44.90	44.97	45.02	44.87	45.23
Crude lipid	10.32	10.17	10.45	10.27	10.51	10.39	10.25
Ash	5.53	5.67	5.60	5.59	5.66	5.62	5.71
Gross energy (MJ/kg)	20.86	21.01	21.11	21.23	20.97	21.20	21.15
Vitamin K_3_ (mg/kg)	0.16	5.25	10.22	14.93	20.51	40.09	59.87

^1^ Provided by Shandong Conqueren Marine Technology Co., Ltd., Weifang, China; ^2^ composition (g/kg mineral premix): AlK(SO_4_)_2_·12H_2_O, 123.7; CuSO_4_.5H_2_O, 32.0; CoCl_2_·6H_2_O, 49.0; FeSO_4_.7H_2_O, 707.0; MgSO_4_.7H_2_O, 4317.0; MnSO_4_.4H_2_O, 31.0; KI, 5.3; NaCl, 4934.0; Na_2_SeO_3_.H_2_O, 3.4; ZnSO_4_.7H_2_O, 177.0; ^3^ vitamin premix supplied the diets with (mg/kg dry diet): retinal palmitate, 0.75; cholecalciferol, 0.04; α-tocopherol, 50.0; thiamine-HCl, 12.0; riboflavin, 30.0; D-calcium pantothenate, 20.0; pyridoxine-HCl, 15.0; choline chloride, 500.0; meso-inositol, 200.0; D-biotin, 0.5; folic acid, 1.5; ascorbic acid, 100.0; niacin, 75.0; cyanocobalamin, 0.01.

**Table 2 biology-14-00447-t002:** Growth performance and feed utilization of coho salmon alevins fed diets with graded levels of VK_3_ for 12 weeks.

Diet Groups	Dietary Vitamin K_3_ Levels (mg/kg)							*p*-Value
	0.16	5.25	10.22	14.93	20.51	40.09	59.87	
SR (%)	95.33 ^c^ ± 0.88	96.00 ^bc^ ± 0.58	96.67 ^abc^ ± 0.88	97.33 ^abc^ ± 0.88	98.33 ^a^ ± 0.33	97.67 ^ab^ ± 0.67	98.33 ^a^ ± 0.33	0.05
IBW (g)	0.32 ± 0.03	0.33 ± 0.04	0.32 ± 0.02	0.32 ± 0.04	0.33 ± 0.03	0.33 ± 0.02	0.33 ± 0.03	0.53
FBW (g)	4.32 ^e^ ± 0.03	4.44 ^d^± 0.04	4.59 ^c^ ± 0.02	4.90 ^a^ ± 0.04	4.87 ^a^ ± 0.03	4.83 ^ab^ ± 0.02	4.76 ^b^ ± 0.03	<0.001
WG (g)	4.00 ^e^ ±0.03	4.12 ^d^ ±0.04	4.27 ^c^ ±0.02	4.58 ^a^ ± 0.04	4.55 ^a^ ±0.03	4.51 ^ab^ ±0.02	4.44 ^b^ ± 0.03	<0.001
SGR (%/d)	3.09 ^f^ ± 0.01	3.11 ^e^± 0.01	3.16 ^d^ ± 0.01	3.24 ^a^ ± 0.01	3.22 ^ab^ ± 0.01	3.21 ^bc^ ± 0.01	3.19 ^c^ ± 0.01	<0.001
FCR (%)	1.27 ^a^ ± 0.02	1.22 ^b^ ± 0.01	1.17 ^c^ ± 0.01	1.08 ^d^ ± 0.01	1.08 ^d^ ± 0.01	1.10 ^d^ ± 0.01	1.11 ^d^ ± 0.01	<0.001
K (%)	1.66 ± 0.05	1.71 ± 0.03	1.62 ± 0.04	1.60 ± 0.04	1.61 ± 0.05	1.65 ± 0.04	1.73 ± 0.03	0.28
HSI (%)	1.16 ± 0.03	1.10 ± 0.04	1.14 ± 0.03	1.08 ± 0.04	1.15 ± 0.07	1.16 ± 0.04	1.11 ± 0.05	0.79
VSI (%)	7.68 ± 0.07	7.86 ± 0.06	7.69 ± 0.09	7.74 ± 0.04	7.78 ± 0.07	7.49 ± 0.11	7.63 ± 0.09	0.09

^a–f^ Within the same row, mean values with different superscripts differ significantly (*p* < 0.05) with mean ± SE. Abbreviations: SR: survival rate; IBW: initial body weight; FBW: final body weight; SGR: specific growth rate; FCR: feed conversion ratio; CF: condition factor; HSI: hepatosomatic index; VSI: viscerosomatic index.

**Table 3 biology-14-00447-t003:** Body muscle proximate composition (%) of coho salmon alevins fed diets with graded levels of VK_3_ for 12 weeks (on a dry matter basis).

Dietary Vitamin K_3_ Levels (mg/kg)	Moisture	Crude Protein	Crude Lipid	Ash
0.16	73.13 ± 0.26	19.65 ± 0.12	3.66 ± 0.12	3.28 ± 0.10
5.25	72.93 ± 0.34	19.72 ± 0.18	3.61 ± 0.15	3.36 ± 0.09
10.22	73.03 ± 0.42	19.74 ± 0.14	3.55 ± 0.15	3.39 ± 0.12
14.93	72.74 ± 0.57	19.55 ± 0.19	3.63 ± 0.10	3.48 ± 0.07
20.51	73.01 ± 0.50	19.39 ± 0.26	3.54 ± 0.11	3.51 ± 0.07
40.09	73.23 ± 0.53	19.58 ± 0.21	3.75 ± 0.09	3.56 ± 0.10
59.87	72.88 ± 0.32	19.53 ± 0.17	3.71 ± 0.11	3.54 ± 0.08
*p*-Value	0.99	0.86	0.86	0.33

**Table 4 biology-14-00447-t004:** Whole-body and liver vitamin K (MK-4) concentrations of coho salmon alevins fed diets with graded levels of VK_3_ for 12 weeks.

Dietary Vitamin K_3_ Levels (mg/kg)	Whole-Body MK-4 Concentration (ng/g)	Liver MK-4 Concentration (ng/g)
0.16	5.14 ^g^ ± 0.17	33.76 ^d^ ± 0.73
5.25	21.97 ^f^ ± 0.39	257.65 ^c^ ± 4.29
10.22	33.58 ^e^ ± 0.74	362.58 ^b^ ± 6.41
14.93	42.99 ^d^ ± 0.70	423.88 ^a^ ± 3.22
20.51	47.65 ^c^ ± 0.41	433.07 ^a^ ± 4.82
40.09	64.12 ^b^ ± 0.73	437.05 ^a^ ± 3.72
59.87	75.24 ^a^ ± 0.95	435.85 ^a^ ± 7.33
*p*-Value	<0.001	<0.001

^a–g^ Within the same column, mean values with different superscripts differ significantly (*p* < 0.05) with mean ± SE.

**Table 5 biology-14-00447-t005:** Liver anti-oxidative enzyme activities of coho salmon (*Oncorhynchus kisutch*) alevins fed diets with graded levels of VK_3_ for 12 weeks.

Dietary Vitamin K_3_ Levels (mg/kg)	T-AOC (U/mg prot.)	T-SOD (U/mg prot.)	CAT (U/mg prot.)	MDA (nmol/mg prot.)
0.16	20.02 ^c^ ± 0.80	118.90 ^d^ ± 5.01	7.38 ^f^ ± 0.10	11.52 ^a^ ± 0.17
5.25	23.34 ^b^ ± 0.54	143.86 ^c^ ± 7.34	9.50 ^e^ ± 0.16	9.46 ^b^ ± 0.14
10.22	26.27 ^a^ ± 0.39	173.77 ^ab^ ± 5.44	12.07 ^b^ ± 0.15	7.71 ^d^ ± 0.15
14.93	28.19 ^a^ ± 0.54	182.22 ^a^ ± 5.95	13.33 ^a^ ± 0.18	6.65 ^e^ ± 0.09
20.51	27.80 ^a^ ± 0.74	176.70 ^ab^ ± 6.05	11.23 ^c^± 0.29	6.80 ^e^ ± 0.10
40.09	26.57 ^a^ ± 0.66	168.30 ^ab^ ± 6.37	10.50 ^d^ ± 0.16	7.35 ^d^ ± 0.12
59.87	23.49 ^b^ ± 0.71	156.68 ^bc^ ± 6.97	10.23 ^d^ ± 0.23	8.63 ^c^ ± 0.11
*p*-Value	<0.001	<0.001	<0.001	<0.001

^a–f^ Within the same column, mean values with different superscripts differ significantly (*p* < 0.05) with mean ± SE. Abbreviations: CAT: catalase; MDA: malondialdehyde; T-AOC: total antioxidant capacity; T-SOD: total superoxide dismutase.

## Data Availability

All the data in the article are available from the corresponding author upon reasonable request.

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
