# Peer review of "Effects of Varying Dietary Concentrations of Menadione Nicotinamide Bisulphite (VK3) on Growth Performance, Muscle Composition, Liver and Muscle Menaquinone-4 Concentration, and Antioxidant Capacities of Coho Salmon (Oncorhynchus kisutch) Alevins"

_biology, 2025, doi:10.3390/biology14040447_

Round 1
Reviewer 1 Report
Comments and Suggestions for Authors
- On what basis was the 12-week experimental period chosen?
- Figures 2 and 3 need to be briefly commented.
Author Response
We sincerely thank the reviewers for their thoughtful comments and constructive suggestions. Their insights have greatly contributed to improving the clarity, quality, and scientific rigor of our manuscript.

Reviewer 2 Report
Comments and Suggestions for Authors
The article entitled " Effects of varying dietary concentrations of Menadione (Vitamin K3) on growth performance, body composition, liver & muscle menaquinone-4 (MK-4) concentration, and antioxidant capacities of coho salmon (Oncorhynchus kisutch) alevins " provides interesting information for researchers who are interested in dietary supplements in aquaculture especially coho salmon. The following comments and suggestions could be considered to improve the quality of the manuscript. 
- Throughout the manuscript, please provide the full term only at its first occurrence and use the same abbreviation consistently for the same term throughout the document.
- Correcting misspellings, using abbreviations and spacing between numbers and units to ensure consistency in the manuscript. For example,
- Different full terms and abbreviations are used to refer to the same thing such as
- Menadione, vitamin K3 (menadione), Menadione K3 (VK3), VK3 versus VK3
- Line 300-302, 316, 317, etc: VK should be replaced by VK3.
- MK4, MK-4
- Line 63, 80: micronutrient, micro-nutrient
- Line 60: farming
- Line 116: alevins
- Line 166: Vitam K4 (MK4) should be changed to menaquinone-4.
- Line 203: mg/L
- Line 289: Abbreviation
- Line 297: Consider replacing “3.3. MK-4 Concentration Determination in Whole Body and Liver Tissue Liver Anti-Oxidative Enzyme Activity” with 3.3. MK-4 Concentrations in Whole Body and Liver Tissue and Liver Anti-Oxidative Enzyme Activity.
- Line 299: recorded
- Line 306 and 310: vitamin K (MK-4)
- Line 311: “MK-4, total antioxidant capacity” should be changed to “MK-4, menaquinone-4”.
- Line 352: 2-12 mg/kg
- Line 374: cod
- Line 408: concentration
- Line 415: Consider changing from “…whole body MK-4, liver MK-4 concentration, and total antioxidant capacity in coho salmon…” to “…whole body MK-4 and liver MK-4 concentrations, and liver total antioxidant capacity in coho salmon…”.
- Consider revising the introduction to make it shorter and more concise.
- Consider the correctness of the equation used to calculate weight gain. Is weight gain equal to final weight (g) minus initial weight(g)?
- Is IBM SPSS 21.0 also used for analyzing polynomial quadratic regression? If not, please specify the software and version used.
- Consider providing additional explanation for the reduction in vitamin K3 content in the formulation when 15 mg/kg and 60 mg/kg of vitamin K3 are added.
- Table 2: Consider using 2-digit values for the IBW.
- Table 2: In the caption, the word “column” should be changed to “row”.
- There is “... of three replicate groups” only in the caption of Table 5, but not in the others.
- In Figure 3-5, the equations and coefficient of determination values should be corrected to use superscript.
- Are there any limitations in this study or areas that should be further investigated? Please specify.
Author Response

(The authors gave the same response as above.)

Reviewer 3 Report
Comments and Suggestions for Authors
Brief summary:
The authors present valuable results from a regression analysis study aiming to determine the optimal vitamin K3 (VK3) inclusions levels in the feed for coho salmon alevins. This work provides novel insights into the dietary VK3 recommendations for coho salmon, and thereby addressing a gap in the current knowledge regarding the VK3's role in improving overall growth and strengthen antioxidant defence in this commercially important species. Through a quadratic regression analysis, the authors identify an optimal dietary VK3 range (31.97 to 43.50 mg/kg) that maximized growth, enhanced body composition, and improved antioxidant capacity in alevins. The primary strength of this work lies in its quantitative approach to dietary VK3 optimization. The findings are significant, as they contribute to advancements in aquaculture nutrition, health and growth performance of this species.
General concept comments:
Overall, the manuscript is scientifically sound, and the experimental design is appropriate for testing the hypothesis. The relevance and knowledge gap are addressed, however, the rationale for exploring VK3 inclusion levels should be clarified more, particularly beyond its general importance for health. Additionally, the manuscript would benefit from greater precision in the use of “VK” and “VK3”, as this distinction is crucial. The choice between these forms should be clearly defined and be consistent (e.g. L77, L85-88, L353). Further, there are issues with spelling, formatting, and English language throughout the manuscript. These areas require revision for clarity and readability. The manuscript title may also benefit from integrating the study’s main conclusions, as the current title does not accurately convey the focus on the dietary recommendations. The introduction could be more concise, particularly section L117-138, which could be shortened by half to focus solely on the most relevant information for the manuscript’s main objectives. While most of the references cited are relevant, some could be updated to include more recent publications, where possible. It's also worth checking if all references are fully appropriate in the context (e.g. L74).
Specific comments:
Abstract:
L23: Consider using "fillet quality" instead of "meat quality" for greater precision.
L30-36: these sentences could be combined to avoid redundancy, as the information is repeated.
Introduction:
L74: The reference to "current scenario of freshwater farming" needs revision for clarity and precision.
L76-77: menadione (VK3) is a synthetic analogue of vitamin K (not vitamin K3).
L85: The term "homeostasis" needs further clarification. Specifically, homeostasis of which aspect or component is being referred to?
L85: The sentence requires revision for clarity, and the scientific context needs to be enhanced, particularly how VK3 is metabolized into the vitamin K cycle and the specific role of MK-4.
L92: Remove either “dietary” or “feed” for clarity. The phrasing of the first part of the sentence could be improved, and "less" should be replaced with "few" to ensure correct usage.
L105: Please specify which species the results cited from reference [30] pertain to.
L110: The sentence is overly long and needs revision. Consider breaking it into shorter sentences. The final part of this sentence (L115) could be moved to avoid confusion, as its current placement disrupts the flow of the argument.
L117: The Latin name Oncorhynchus kisutch should be included the first time the species coho salmon is mentioned in the text.
Figure 1 and 2: Both figures appear unnecessary for this scientific work, as they do not provide novel information. I recommend removing them.
Methods:
L217: The term "fish whole body muscle samples" may lead to confusion and needs explanation for the reader. "Whole body" typically refers to analyzing the entire alevin, but if I understood right, you sampled the complete muscle as proxy for whole body. Please clarify this distinction when describing your sample matrix or tissue type throughout the manuscript, including in lines L242-243.
Results:
L285: It would be more appropriate to report the predicted VK3 levels in mg/kg throughout the manuscript, rather than as a percentage, unless there is a clear justification and explanation for using percentage terms. Reporting the VK3 concentrations as a percentage could lead to confusion, as the percentage may not be immediately clear in terms of its reference point. Additionally, 0.16 mg/kg is significantly smaller than 0.16% in the diet. This suggestion applies to units in figures 2-5 and the conclusions in L416.
Discussion:
L352-353. Are Grisdale-Helland et al. suggesting not to use VK in general or the synthetic form of VK3? Please clarify this point.
L364-365. Please provide the correct Latin name for Atlantic salmon (Salmo salar).
L380-383: This sentence should be revised for clarity to make the statement more explicit.
L390-396: Consider splitting it into two sentences and revise the language for greater clarity.
Comments on the Quality of English LanguageThe language and spelling, particularly in the introduction and discussion, require revision for more scientific clarity and readability.
Author Response

(The authors gave the same response as above.)

Reviewer 4 Report
Comments and Suggestions for Authors
The topic of your research is interesting, even if it is not the first study of its kind. I have made some remarks and comments regarding your research. I would like you to comment on them and, if is necessary, correct them. All the best!

Author Response

(The authors gave the same response as above.)

Round 2
Reviewer 3 Report
Comments and Suggestions for Authors
Generally, this work has shown improvement, and the majority of the previous comments have been addressed by the authors. However, I still recommend further attention to the references and the conclusions drawn by the authors. Additionally, as synthetic vitamin K is added to fish feed in various forms (salts), I suggest specifying which menadione salt was used in this work as an additive to the diets (please add in L170).
L110: The Latin name should be introduced earlier in the text (e.g., L105).
L303-304: The phrase "total antioxidant capacity" should be removed, as it appears to be a copy-paste error.
L343-346: The content of this sentence is not accurate. Please ensure the discussion is relevant and the information provided is correct. Specifically, vitamin A does not fit into this context. It is important to clarify that synthetic vitamin K is available in different forms, and while MSB is not recommended, MNB is considered more suitable. Grisdale-Helland et al. have reported on MSB. Please also see this paper on MNB in Atlantic salmon feed: https://doi.org/10.1111/j.1365-2095.2008.00633.x.
L375: Does your data and analysis support the claim that dietary VK3 has been "effectively" converted to MK-4? Please provide further clarification or justification.
L379: Kindly state the doses that were suggested as toxic. Additionally, it may be helpful to specify which synthetic form of vitamin K this refers to.
L412: The recommendation level of VK3 should also here be presented in mg/kg.
L567: There is an error in the correction of the original title. Please revise accordingly.
Author Response

(The authors gave the same response as above.)

Reviewer 4 Report
Comments and Suggestions for Authors
The manuscript has improved based on reviewer suggestion.
Author Response

(The authors gave the same response as above.)
